# The *Alphavirus* Sindbis Infects Enteroendocrine Cells in the Midgut of *Aedes aegypti*

**DOI:** 10.3390/v12080848

**Published:** 2020-08-04

**Authors:** Yani P. Ahearn, Jason J. Saredy, Doria F. Bowers

**Affiliations:** 1Department of Health, TB Lab, 1217 N Pearl St., Jacksonville, FL 32202, USA; yaniyw@gmail.com; 2Department of Biology, Temple University, 1900 N 12th St., Philadelphia, PA 19122-6078, USA; tug39776@temple.edu; 3Department of Biology, University of North Florida, Jacksonville, FL 32224, USA

**Keywords:** *Alphavirus*, enterocytes, enteroendocrine cells, GFP, FMRFamide

## Abstract

Transit of the arthropod-borne-virus (arbovirus) Sindbis (SINV) throughout adult female mosquitoes initiates with its attachment to the gut lumen, entry and amplification in midgut cells, followed by dissemination into the hemolymph. Free-mated adult females, aged day 5–7, were proffered a viremic blood suspension via sausage casings containing SINV-TaV-Green Fluorescent Protein (GFP) at a final titer of 10^6^ PFU/mL. Midguts (MGs) from fully engorged mosquitoes were resected on days 5 and 7 post-bloodmeal, and immunolabeled using FMRFamide antibody against enteroendocrine cells (ECs) with a TX-Red secondary antibody. Following immunolabeling, the organs were investigated via laser confocal microscopy to identify the distribution of GFP and TX-Red. Infection using this reporter virus was observed as multiple GFP expression foci along the posterior midgut (PMG) epithelium and ECs were observed as TX-Red labeled cells scattered along the entire length of the MG. Our results demonstrated that SINVGFP did infect ECs, as indicated by the overlapping GFP and TX-Red channels shown as yellow in merged images. We propose that ECs may be involved in the SINV infection pathway in the mosquito MG. Due to the unique role that ECs have in the exocytosis of secretory granules from the MG and the apical-basolateral position of ECs in the PMG monolayer, we speculate that these cells may assist as a mechanism for arboviruses to cross the gut barriers. These findings suggest that MG ECs are involved in arbovirus infection of the invertebrate host.

## 1. Introduction

Arthropod-borne-viruses (arboviruses), the etiologic agents of many vector-borne diseases, are transmitted by hematophagous insects such as mosquitoes, ticks and midges [1,2]. These arboviruses pose a major health burden worldwide and since the beginning of the 20th century, disease incidence and associated deaths have been on the rise due to an increasing vector habitat range associated with global warming [3,4]. The *Alphaviruses* Chikungunya, O’nyong nyong, as well as Eastern, Western, and Venezuelan equine encephalitis viruses, can cause fatal infections in both humans and lower vertebrates [5,6]. While Sindbis virus (SINV) causes a significant febrile illness, other alphaviruses can result in arthritogenic or encephalic symptoms. SINV is the prototype alphavirus, and these findings may have bearing on other more virulent viruses. The mosquito *Aedes* (*Ae.*) *aegypti*, the Yellow Fever mosquito, well-known as an arbovirus-transmitting vector, is a vector of the *Alphavirus* Sindbis in the laboratory setting [7,8].

SINV, first isolated in 1952 from mosquitoes in Sindbis, Egypt [9], is a zoonotic virus commonly distributed across several continents including Eurasia, Africa, Oceania and Australia [10]. SINV is a membrane bounded virus that has a plus sense, single-stranded RNA genome consisting of ~11,703 nucleotides, a methylated cap on the 5′ end, and a 70-nucleotide long poly-A tail on the 3′ end [11]. Its genome is housed in an icosahedral capsid consisting of 240 capsid monomers and its host-derived lipid membrane is studded with 240 trimeric spikes composed of heterodimers of E1 and E2 glycoproteins, which aid in its attachment to host cells [12]. SINV genome consists of two open reading frames that code for mRNA which subsequently generates four non-structural proteins and five structural proteins. Once SINV enters host cells, translation and assembly of the nucleocapsid occurs in the cytosol, and the completion of the virus assembly occurs at the host cell plasma membrane resulting in the budding of progeny virus [13,14].

Establishment of an arbovirus infection within a mosquito requires that the virus gain access into the midgut (MG) cells to amplify before disseminating into the hemolymph [15]. Barriers to MG infection have been documented, including MG infection barrier and MG escape barrier [16]. Such barriers are integral to permissive and/or refractory nature of infections within the MG epithelium, permitting or preventing secondary infections throughout the whole mosquito [17], all prior to approaching and attaching to the salivary glands for potential virus transmission [18]. The mosquito alimentary canal is composed of a tubular thoracic region and three distinct abdominal regions, the foregut, midgut and hindgut [19]. The posterior midgut (PMG) where blood digestion occurs is further divided into three distinct regions, each approximately a third of the MG: PMG-frontal (PMG-f), PMG-middle (PMG-m), and PMG-caudal (PMG-c) [20]. The MG is a monolayer composed of simple columnar digestive enterocytes and scattered enteroendocrine cells (ECs), all surrounded by a basement membrane, nerve fibers, muscle bundles and tracheoles [21,22]. The columnar enterocytes are heavily decorated by microvilli on the apical aspect, whereas only the open ECs have apical microvilli, a characteristic that is absent on closed ECs [23]. Brown and colleagues [23] estimated that 500 of these basally, solitarily positioned ECs, each ~2–6 μm in diameter were observed to possess either apical or basolateral extensions or both. Dark or clear cytoplasm is observed among these ECs and the secretory granules are stored along the basolateral membrane of the MG.

Upon blood digestion, these secretory granules are released into the extracellular spaces following membrane fusion, ultimately emptying contents via exocytosis [24]. A variety of neuropeptides are also secreted by ECs into the alimentary canal of the mosquito, including phenylalanine-methionine-arginine-phenylalanine-amide (FMRFamide) [24,25,26]. FMRFamide was the first cardioexcitory peptide identified in mollusks and is distributed exclusively in mosquito PMG region where the blood is deposited [27,28]. SINV infects and replicates in the host species *Ae. aegypti* and in this study, the prototype *Alphavirus* SINV, was used to gain insight into the biology of alphaviruses. We used a reporter virus (SINV-TaV-GFP) that contains a GFP genomic insert between the capsid protein and E2, maintaining the capsid auto protease on the 3′ end and *Thosea asigna* virus (TaV) 2A-like protease at the 5′ end [29]. GFP protein products are cleaved upon virus structural protein synthesis while the rest of the virus remains intact, thereby allowing us to track the sites of virus infection without any physical changes to the virion. As SINV-TaV-GFP replicates in the MG cells of mosquitoes, a GFP trail is retained in these cells while continuing to infect other target tissues. Previously, our lab documented that SINV-TaV-GFP foci in the mosquito MG were first detected at day 3 post-infection (p.i.), located in all three regions of PMG, and most MGs revealed 1 focus, or 2–3 foci in *Ae. aegypti* [20]. An MG focus is likened to a cell culture virus plaque in that an initial cell is infected and the focus expands outwards by cell-to-cell spread of progeny virions while the number of plaques represents the virus titer or in this case, the number of MG cells initially infected. This research has identified SINV-TaV-GFP accumulations in MG ECs, confirming infection of these MG cells in vivo. Considering the unique location and function of MCs, we hypothesize that SINV uses the secretary nature of these cells to release virus into the hemolymph. We seek to determine if an arbovirus is capable of an infection of MG ECs in adult female *Ae. aegypti* mosquitoes following a viremic bloodmeal.

## 2. Results

### 2.1. Distribution of SINV MG Infection Foci

Female *Ae. aegypti* mosquitoes were challenged with a viremic bloodmeal at days 5–7 post-emergence. Whole-mount MGs were resected into glass-bottomed cell culture dishes and observed for SINV-associated GFP fluorescence via an Olympus laser scanning confocal microscope. GFP infection foci were clearly and distinctly visible in all three regions of the PMG (Figure 1A,B).

### 2.2. Distribution of FMRFamide Positive ECs

Morphological evidence of enterocytes and ECs surrounding the MG lumen were observed in cross-section of the mosquito MG monolayer (Figure 2A,B). This organ was lined with microvilli on the apical aspect, enveloped with a basal lamina surrounding lighter-stained enterocytes interspersed with darker-stained ECs. ECs appear conical, with a flask-like shape communicating with apical or basal or both aspects of the MG. This morphology of ECs reflects reports [23,26] of ECs located in PMG. Antibody against FMRFamide was used for the detection of ECs positive for FMRFamide in the mosquito PMG and visualized via a secondary antibody conjugated with TX-Red fluorochrome (Figure 3). Spatial distribution of ECs was observed along the entire length of the PMG.

### 2.3. GFP Virus Infection Foci and FMRFamide-Positive ECs

Female *Ae. aegypti* were blood-fed with SINV-TaV-GFP bloodmeals, followed by the identification of ECs positive for FMRFamide-TX-Red labeling (Figure 4A). GFP accumulations sequestered in gut cells were evident at day 5 p.i. forming a single infection focus in the epithelia of the PMG-m, observed in two different rotated views (Figure 4B,C). Overlay of the GFP accumulations (Figure 4B) with FMRFamide-TX-Red positive ECs (Figure 4A) results in a merged image (Figure 4D), which demonstrated that colocalization of arbovirus and ECs was not observed at day 5 p.i. At day 5 p.i., a total of 60 mosquitoes imbibed viremic blood, virus foci were identified in 4 mosquitoes (~7% infection) and colocalization with ECs was absent.

Further experiments examining the distribution of GFP accumulations and FMRFamide labeling demonstrated colocalization of SINV infection foci and ECs in the MG of a female mosquito on day 7 p.i. Confocal analysis of the MG demonstrated GFP accumulations in gut cells forming a single infection focus in the PMG region (Figure 5A) and the presence of FMRFamide positive ECs (Figure 5B). Overlay of the GFP accumulations and FMRFamide positive ECs demonstrated colocalization of the two (shown in yellow; Figure 5C) in mosquito #1. This phenomenon was repeated in additional experiments with mosquitoes # 2 (Figure 6A–C), #3 (Figure 7A–C) and #4 (Figure 8A–C) showing a yellow color where concentrations of SINV-GFP in the foci and ECs TX-Red colocalized at distinctive sites in the PMG (Figure 5, Figure 6, Figure 7 and Figure 8). A total of 215 mosquitoes imbibed viremic blood, virus foci were identified in 75 mosquitoes (~35% infection) and colocalization of foci and ECs was observed in four individual mosquitoes (~5%) at day 7 p.i. (Table 1).

### 2.4. Percent Infection of Mosquito MGs at Day 7 p.i.

Percent of mosquitoes infected with SINV foci distribution in the MG was determined using confocal microscopy (Table 1). The presence or absence of SINV associated GFP foci are indicative of permissive (infected) vs refractory (noninfected) mosquitoes. Out of 215 mosquitoes surveyed, 35% of the mosquitoes were permissive to SINV-TaV-GFP, making it an in vivo infectious arbovirus. Note that the majority of SINV-TaV-GFP foci were observed in the PMG-m while the rest of GFP foci were observed in lesser numbers in PMG-f and PMG-c in agreement with previous research conducted in our lab [20].

### 2.5. Multiple Infection Foci in Ae. aegypti MGs

Numerous infection foci (>12) were observed throughout all three segments of the MG of a female mosquito on day 7 p.i. (Figure 9). This finding was an outlier from all other mosquitoes utilized in our research because of the high number of foci and the finding may underscore physiological or genetic differences.

## 3. Discussion

Whether MG ECs have a role in SINV infection of the mosquito host has been a long-time topic of discussion. Using the GFP-reporter virus enabled us to map the temporal–spatial timeline of infection of MG cells, and SINV infection foci were ubiquitously expressed in all three distinct regions of the PMG on both day 5 and day 7 p.i. While SINV dissemination to peristaltic muscles was not detected on neither day 5 nor day 7 p.i., this could be attributed to the sample size of the current study as well as individual physiological differences in the same mosquito species. Interestingly, we discovered a female mosquito MG infected with SINV displaying multiple GFP foci (>12) at day 7 p.i., an unusual finding considering that the usual number of GFP foci is 1–3 from our hands [20], or the 6 foci observed in Figure 1. Numerous virus foci highlighted the individuality between mosquitoes of the same species and may indicate high transmitters. This, too, demonstrates that individual differences are significant in each mosquito and need to be considered when conducting experiments.

Previous documentation indicates that ECs are the most abundant in the PMG of female mosquitoes where blood digestion takes place [20] and our data supports this in *Ae. aegypti*. Immunopositive ECs and GFP foci did not colocalize at day 5 p.i. (Figure 4) but did colocalized at distinct sites in the PMG on day 7 p.i. (Figure 5, Figure 6, Figure 7 and Figure 8). This finding indicates that ECs are permissive and involved in SINV MG infection. Further research is needed to assay additional days p.i. for colocalization of SINV and ECs, and we are specifically interested in investigating days 2–4 p.i. Upon bloodmeal ingestion, FMRFamide is released from the basolateral aspect of the gut into the hemolymph, where it acts as a paracrine signal to transduce downstream signaling to nearby and distant tissues [26]. These researchers suggested that stretching of the gut due to the presence of bloodmeal may aid in the release of FMRFamide. Given its cardioexcitory and myotropic nature, FMRFamide could then stimulate the contraction of the gut peristaltic muscles to move digested bolus bloodmeal past the PMG to the hindgut for excretion [27]. Additionally, FMRFamide could function as a hormone that diffuses to neighboring digestive enterocytes to stimulate the release of an appropriate number of digestive enzymes to prevent self-digestion [26]. Interestingly, relevant sensory cues control secretions of FMRFamide-related neuropeptides in nematodes, where they act directly on the egg-laying motor neurons, thereby directly modulating their reproductive behaviors such as egg-laying and copulation [30,31]. Given the abundance and physiological complexity of ECs, we suggest that in addition to enterocytes, SINV can use ECs to initiate infection and subsequently hijack the unique secretory function of the cells for virus dissemination from the MG epithelia.

This investigation demonstrated a novel finding that SINV infects ECs, indicating that these cells are permissive to SINV and may indicate the presence of virus receptors [32] on EC membranes. Hence, this alphavirus could potentially utilize the neurosecretory nature of the ECs to disseminate to neighboring MG cells and distant target tissues. Our findings contributed to the current understanding of the underlying mechanisms that arboviruses utilize to overcome infection barriers in their insect hosts. However, our study does not negate other potential avenues arboviruses could exploit for establishing successful and persistent infection in their mosquito hosts. Colocalization of SINV and EC was not observed at day 5 p.i. and 5.3% colocalization of SINV in ECs was identified on day 7 p.i. This colocalization was observed in one EC in 4 different mosquitoes. Quite possibly, this reflects the overall low infection rate (7% on day 5 p.i.; 35% on day 7 p.i.) observed. It maybe that our study just missed the merged yellow color on the MG because it can be difficult to manipulate the gut in order to observe the backside of a whole-mount preparation. Alternatively, the MG EC cells are possibly more physiologically receptive earlier following a bloodmeal prior to or during the deposition of the peritrophic membrane [19]; or quite simply, dissection trauma or loss of gut tissues can hamper quantification of our research. While the percent of infection of MGs is not high, it only takes a few infected mosquitoes to spread an arbovirus in nature. Detection of chikungunya virus in the saliva of *Ae. aegypti* as early as day 2 p.i. [33] is encouraging, and we plan to expand our assay to include days 2–4 p.i. during the acute phase of infection [17]. Due to global warming, disease-carrying insects will spread to areas where arbovirus outbreaks are not typically a concern, and it is crucial to gain more understanding of arbovirus biology, transmission and host response to infection to combat future infectious disease outbreaks.

## 4. Materials and Methods

### 4.1. Hatching and Maintenance of Colony Mosquitoes

*Aedes aegypti* mosquitoes (USDA, Gainesville, FL, USA) were reared and arbovirus experimentation using adults was conducted in the UNF BSL-2 insectary under standard environmental conditions (25.5 ± 0.5 °C, 70–80% humidity, lighting with 30 min gradual brightening and 30 min of gradual darkening bracketing a 16:8 light/dark photoperiod) [34]. Mosquito eggs were hatched in 1.0% nutrient broth (Becton Dickson Microbiology Systems, Spanks, MD, USA). First instar larvae were distributed approximately 300/rearing pan in 1.5 L tap water and fed a 2% liver power suspension (ICN Biochemicals, Cleveland, OH, USA). Following pupation, adults emerged into plastic cages supplied with water-soaked cotton balls for hydration and honey-soaked cellucotton on top of mosquito netting as a carbohydrate source.

Adult female mosquitoes, aged 5–7 days, were supplied with 10 mL of warmed defibrinated bovine blood (Colorado Serum Company, Denver, CO, USA) in collagen sausage casings (22 mm; The Sausage Maker Inc., Buffalo, NY, USA) as a protein source for egg production. Blood-filled sausage casings were hung vertically [35] in gallon-sized mosquito rearing buckets for an hour at standard insectary conditions. Oviposition cups lined with filter paper were supplied after each blood meal and eggs were collected approximately 3–5 days after feeding. Eggs were stored in a humidity chamber under standard conditions to ensure proper embryogenesis and following hatching, female mosquitoes aged 5–7 days post-emergence were used for all experiments.

### 4.2. Morphology of Gut Cross-Section

Midguts were resected from *Ae. aegypti* and split into anterior MG, PMG and hindgut. Tissue was then fixed for 48 h with 2% glutaraldehyde and 0.1% tannic acid in 0.1M cacodylate buffer (pH 7.4) at 4 °C. A secondary fixation with 1.0% OsO_4_ for 30 min was applied, then dehydrated via increasing (30%, 50%, 70%, 95%, 100%) concentrations of ETOH washes, and then placed into propylene oxide transitional fluid. Samples were then embedded into Epon 812 resin with a 3:7 ratio of NMA:DDSA for a slightly softer than medium hardness of resin [36]. Sections were cut at 1–2 μm and stained with 0.1% methylene blue (pH 11) for 90 s on a hotplate at medium temperature (Fisherscientific, part of Thermo Fisher Scientific; all chemicals above in 4.2). Tissues were observed with an Olympus FluoView Laser Scanning Biological Microscope (FV1000 IX81 confocal microscope) using photomultiplier tube transmitted light detector.

### 4.3. Virus Growth and Plaque Assay

SINV-TaV-GFP [29] were donated by the Klimstra Lab (University Pittsburgh) or grown in BHK-21 (baby hamster kidney) cells. These cells were grown in 25 cm^2^ cell flasks at 37 °C, 5% CO_2_ in minimum essential media (MEM) enriched with 5% FBS (ATCC), 5% TPB, and 20 μL gentamicin (Fisherscientific). BHK-21 cells were grown to pre-confluency, 200 uL dilution of virus was adsorbed on cells for 1 h, washed with PBS-D and replaced with 3 mL of EMEM. Virus was grown for 24 h, media was harvested, spun-down at 2000 rpm for 10 min, and supernatant containing extracellular virus was titered via plaque assay. Supernatant of virus was aliquoted and stored at −80 °C until needed. A double overlay agarose assay stained with neutral red was used to quantify virus titer by plaque assay [17,37]. Monolayer of BHK cells were grown to pre-confluency, and ten-fold serial dilutions of SINV/virus growth media (SINV/VGM; 3% fetal bovine serum in PBS) were adsorbed on cell flasks for 1 h with a slow constant rocking on a rocker and periodical rocking by hand. The inoculum was discarded and replaced with 7.5 mL media (1:1 mixture of 2% agarose (Fischerscientific) and 2× MEM) warmed to 42 °C was added to each flask. After the agarose mixture was solidified at RT, the cell flasks were incubated at 37 °C for two days. Once plaques were visible, the agarose layer was overlaid with 5 mL neutral red mixture consisting of (3% neutral red and 1% agarose in PBS-D). Visible plaques were counted at 48 h incubation and a final virus titer was 2.5 × 10^7^ PFU/mL.

### 4.4. Spatial Distribution of SINV Foci and Enteroendocrine Cells in MGs

Mosquito cohorts of 50 were used to examine the distribution of SINV foci and ECs in the PMGs. Mosquitoes were carbohydrate-starved for 24 h prior to blood-feeding. Infectious blood meals were prepared to deliver a final titer of 2.5 × 10^6^ PFU/mL blood. One mL of Sindbis virus (SINV TR339-TaV-GFP) was added to 9 mL of warmed bovine blood contained in a sausage casing. Females replete with blood were gently moved into labeled cages for incubation at standard insectary conditions and MGs were resected on days 5 and 7 p.i., while viewed through a dissection microscope (Leica Microsystems). Resected MGs were transferred into a glass-bottom cell culture dish (Greiner Bio-One) and incubated in 4% paraformaldehyde/PBS for 10 min followed by PBS washes at RT.

### 4.5. Immunofluorescent Labeling of Mosquito Tissues

To detect EC cells, fixed whole-mount MGs were rinsed with PBS and incubated with 10% normal goat serum (NGS)/for 1 h at RT. Treated tissues were incubated with anti-FMRFamide primary antibody (Immunostar CAT# AB_572232) at a dilution of 1:100 in PBS, followed by gentle rocking overnight at RT. Primary antibody was washed with PBS and tissues were treated with goat-anti-rabbit TX-Red secondary antibody (Invitrogen CAT# T-2767) at a 1:20 dilution in PBS for 2 h with gentle rocking at RT. The specificity of the secondary antibody was ensured by incorporating controls with primary antibody only and secondary antibody only in this experiment. Tissues were observed with an Olympus FluoView, FV1000 laser confocal microscopy at 594 nm wavelengths and spatial localization of GFP-SINV and distribution of TX-Red labeled EC’s were photographed and analyzed using ImageJ (NIH).

## Figures and Tables

**Figure 1 viruses-12-00848-f001:**
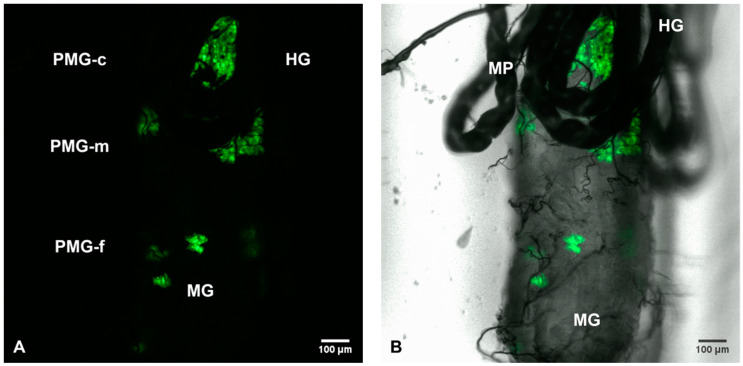
(**A**) Fluorescent laser confocal image and (**B**) bright field overlay of multiple SINV-TaV-GFP infection foci in the MG of *Ae. aegypti* on day 5 p.i. following a viremic bloodmeal. Six SINV infection foci are observed as green fluorescent clusters. Posterior midgut (PMG) regions identified are (**A**) PMG-frontal (PMG-f), PMG-middle (PMG-m) and PMG-caudal (PMG-c). HG, hindgut; MP, Malpighian tubules; MG, midgut. 100×.

**Figure 2 viruses-12-00848-f002:**
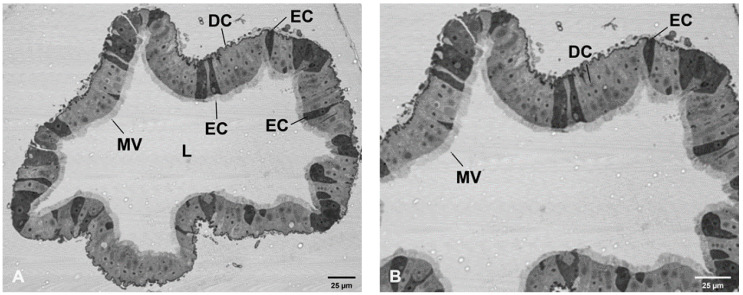
Bright field image of *Aedes aegypti* MG cross-section. Morphological evidence of enteroendocrine cells (EC) integral to the MG monolayer surrounding the lumen (L) in the mosquito PMG. ECs have a darker appearance, have a conical, flask-like shape, unlike neighboring lighter appearing digestive cells (DC). MV, microvilli. (**A**) 100× (**B**) 200×.

**Figure 3 viruses-12-00848-f003:**
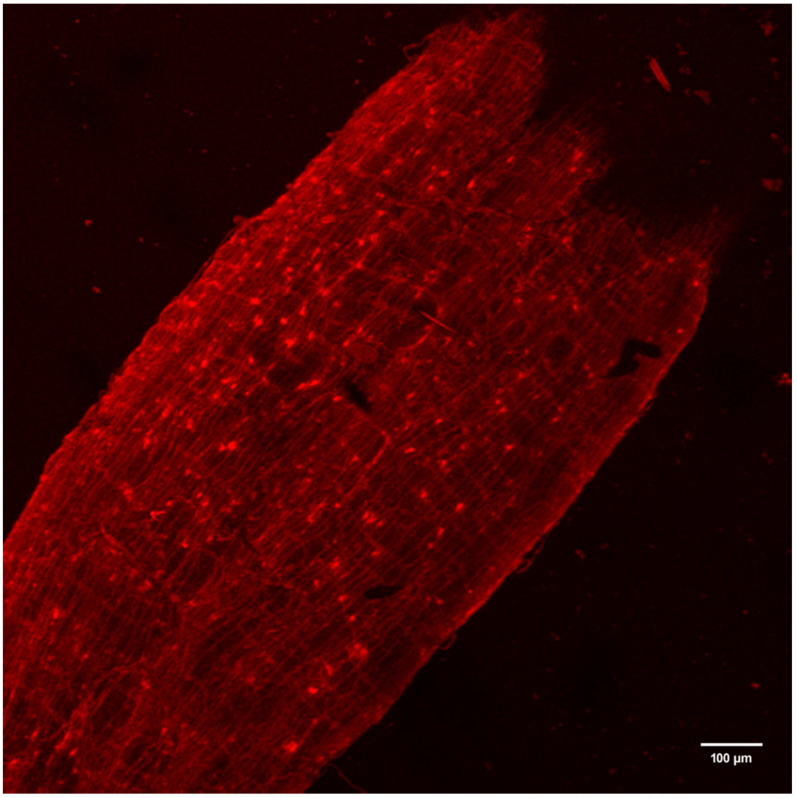
Fluorescent laser confocal micrograph showing the distribution of ECs in the MG from a non-viremic blood-fed mosquito at day 5 post-bloodmeal. Immunoreactive ECs labeled for FMRFamide granules are observed along the entire length of the PMG demonstrated by cell-associated TX-Red fluorescence. 200×.

**Figure 4 viruses-12-00848-f004:**
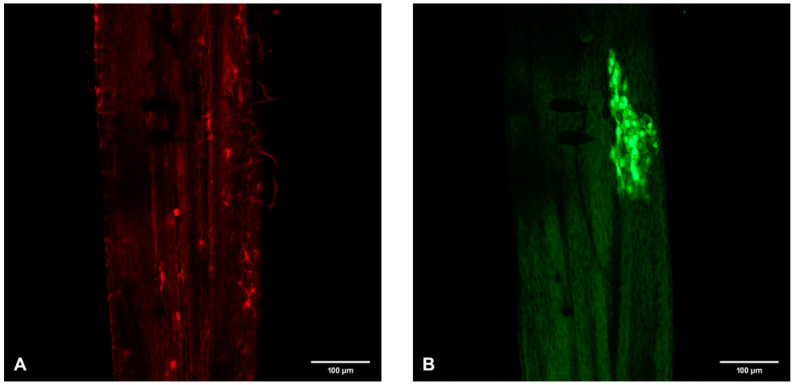
(**A**) Fluorescent laser confocal images of TX-Red FMRFamide positive ECs and (**B**,**C**) sequential images of a SINV-GFP (green) foci extending from front to back of the MG. (**D**) Overlay image of the GFP infection foci (B) and TX-red stained EC image (A) results in merged image. Midgut of a female *Ae. aegypti* dissected on day 5 p.i. and GFP SINV infection focus did not colocalize with FMRFamide-positive ECs. 200×.

**Figure 5 viruses-12-00848-f005:**
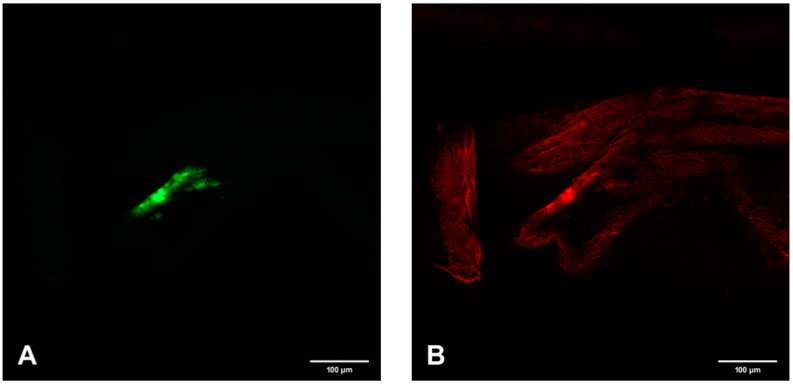
Confocal microscopic analysis of SINV-associated GFP (green) and EC specific FMRFamide-TX Red immunoreactivity in the PMG-m of mosquito #1 at day 7 p.i. (**A**) SINV-associated GFP accumulations in the PMG. (**B**) FMRFamide-positive ECs in the PMG region. (**C**) Bright-field background and overlay of (**A**,**B**) demonstrating the yellow color observed when SINV and EC colocalize. 200×.

**Figure 6 viruses-12-00848-f006:**
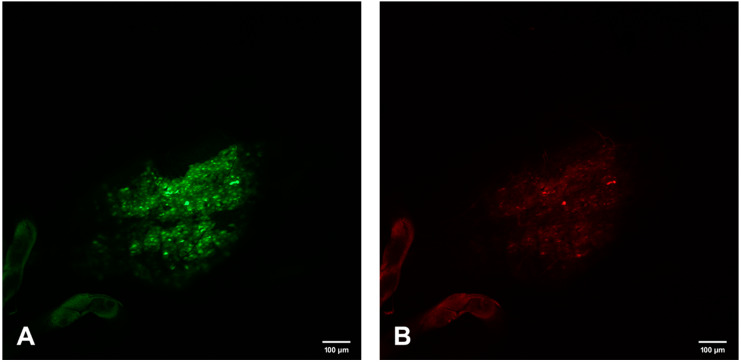
Confocal microscopic analysis of SINV-GFP accumulations and EC specific FMRFamide-TX Red immunoreactivity in PMG-m of mosquito #2 on day 7 p.i. (**A**) GFP accumulations indicating SINV replication. (**B**) FMRFamide immunoreactive ECs. (**C**) Bright-field background and overlay of (**A**,**B**) demonstrating the yellow color (white arrows) observed when SINV and EC colocalize. 200×.

**Figure 7 viruses-12-00848-f007:**
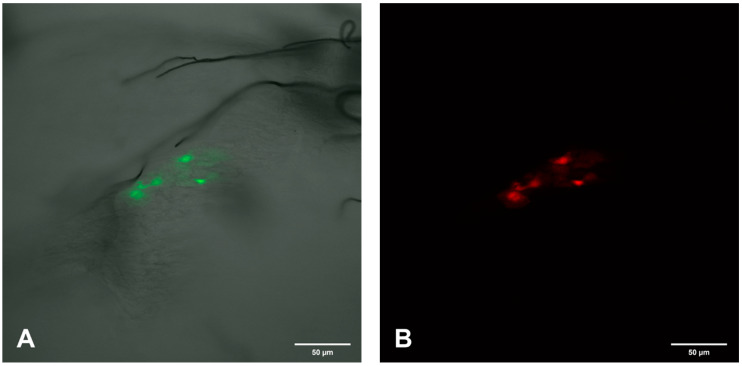
Confocal microscopic analysis of SINV-GFP (green) infection focus and EC specific FMRFamide-TX Red immunoreactivity in the PMG region of mosquito #3 day 7 p.i. (**A**) Single GFP infection focus. (**B**) FMRFamide positive ECs. (**C**) Merged image of (**A**,**B**) demonstrating the yellow color observed when SINV and EC colocalize. 200×.

**Figure 8 viruses-12-00848-f008:**
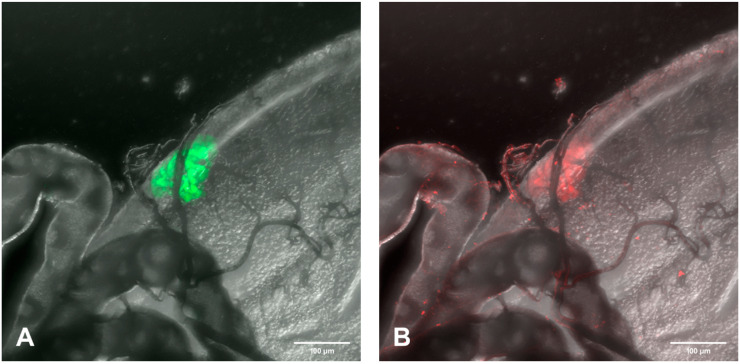
Higher magnification of fluorescent laser confocal analysis of SINV-GFP (green) infection focus and EC specific FMRFamide- TX Red immunoreactivity in the PMG region of mosquito #4 at day 7 p.i. (**A**) Single GFP infection focus. (**B**) FMRFamide positive ECs. (**C**) Merged image of (**A**,**B**) demonstrating the yellow color observed when SINV and EC colocalize. 400×.

**Figure 9 viruses-12-00848-f009:**
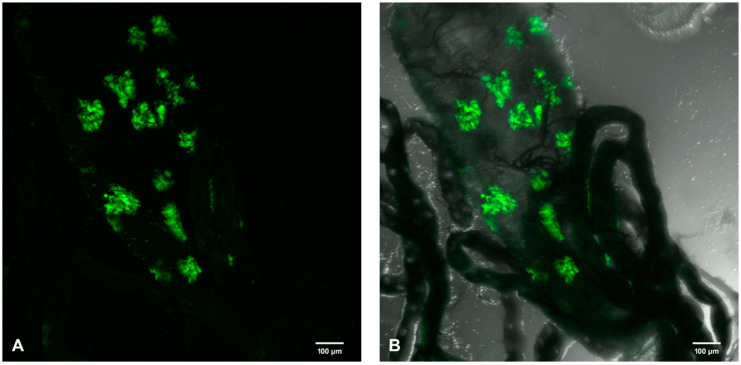
Multiple GFP (green) foci in PMG region of female mosquito D7 p.i. Female mosquitoes were proffered viremic bloodmeal and MGs were dissected on D7 p.i. Unusually high number (>12) of GFP foci (**A**,**B**) in an individual MG are indicative of multiple sites of SINV infection within the mosquito MG. 200×.

**Table 1 viruses-12-00848-t001:** Percent infection of mosquito MGs and distribution of SINV-associated GFP foci in midguts at day 7 p.i.

	Mosquito Infected	PMG-f	PMG-m	PMG-c
**Trial 1**	36/76 = 47%	14/67 = 21%	42/67 = 63%	11/67 = 16%
**Trial 2**	11/20 = 55%	5/15 = 33%	7/15 = 47%	3/15 = 20%
**Trial 3**	6/41 = 15%	4/9 = 44%	2/9 = 22%	3/9 = 33%
**Trial 4**	22/78 = 28%	9/21 = 42%	6/21 = 29%	6/21 = 29%
**Total**	75/215 = 35%	32/112 = 29%	57/112 = 51%	23/112 = 21%

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
