# Peer review of "The Alphavirus Sindbis Infects Enteroendocrine Cells in the Midgut of Aedes aegypti"

_viruses, 2020, doi:10.3390/v12080848_

Round 1

Reviewer 1 Report

in the manuscript Ahearn et al. describe the confocal imaging of GFP expressing cells in Ae aegypti MG after GFP-SINV reporter virus infection and colocalisation with FMRFamide positive EC. 

Major comments:

The authors conclude that no EC are infected at day 5, as opposed to day 7 pi. 

However, it is unclear how many mosquitos were assayed for colocalisation at day 5.

In case only limited mosquitos were assessed this could be due to chance, and would not allow to conclude that early in infection these cells are not targeted. 

The authors should clarify how many mosquitos were assessed at day 5. 

At day 7 pi, the authors show GFP foci with overlayed FMRFamide positive ECs in several mosquitos at high amplification. While this allows examination of colocalisation, it does not allow to judge the presence and overlap of other GFP foci. The authors should quantify how often the GFP foci overlap with ECs. 

minor comments:

line 37: fatal infections are exceedingly rare with these alphaviruses, perhaps the authors can stress that indeed these infections more often cause a significant febrile ilness and depending on the virus arthritogenic or encephalitic symptoms. 

line 42: authors refer to SINV as a cubic virus. It is unclear what the authors refer to in terms of viral structure. Perhaps the capsid organization. However, it is hard to find a  contemporary reference to the description of SINV as having a cubic organization. 

Author Response

We thank all Reviewers for your thoughtful comments. We have thoroughly discussed these comments, made changes and believe that we now have a much improved manuscript. We have addressed all comments to the best of our ability and hope for your approval. Bowers' Lab. Questions regarding coloclization on Days 5&7 p.i. Lines 131-133 indicates the numbers for day 5p.i. and Lines 142-144 indicates the numbers for day 7 p.i. This is further explained in Discussion, L 229-239. Plan to look at day 2-4 p.i. While day 2 maybe difficult because blood still exists in the gut at 48 hour post bloodfed, we will try our best. Day 7p.i.high mag images Lines 230-231, Images were taken at this high mag to underscore the merges colocalization of GFP and TX-Red. Because we did not detect many, there was not a need to image at low mag. Lines 41-43, Pathology of Alphaviruses. Line 48, Cubic virus changed to membrane bounded and icosahedral.

Reviewer 2 Report

The present manuscript by Ahearn et al. ‘The Alphavirus Sindbis infects enteroendocrine cells in the midgut of Aedes aegypti’ shows predominantly fluorescent microscopy images of mosquito midguts to demonstrate that enteroendocrine cells (ECs) in the mosquito midgut can be infected by Sindbis virus. While overall a welcome study, it is not particularly detailed in its analysis, relies heavily on individual confocal images, and thus some of the conclusions drawn from it cannot be supported by the data that’s presented. The main conclusion that can be made from this manuscript is that ECs can be infected (i.e. the title is fine, but parts of the abstract and discussion make claims beyond this). I think the data are useful and interesting, but the way they are presented and the conclusions drawn from the data should be tidied up.

My main concern is thus with the conclusions drawn from the data, e.g. in abstract (line 23-28). The authors talk about the potential relevance for ECs during dissemination of SINV from the midgut. However, the presented data only shows co-localization of SINV and ECs on day 7 post infection. The only earlier sample, from day 5, does not show co-localization. By day 7, my strong assumption is that all infected mosquitoes will have established a disseminated infection, making infection of ECs at that point irrelevant to dissemination. However, the authors don’t even provide any data on dissemination that may enable interpretation of this. Further, it is possible that ECs were infected earlier and the shown difference between day 5 and day 7 is just coincidence, but the authors did not provide sufficient data to show any role for ECs in dissemination. They merely show that at some point during infection, ECs can be infected with SINV. It would have been significantly more beneficial to take early time-points, such as 12, 24, 48 hours post infection and also measure virus dissemination in other tissues, if the intent was to try and connect EC infection to a role in dissemination. From the provided data, these conclusions are not backed up at all.

Minor comments:

Line 23: this may be a bit pedantic, but it’s not ‘yellow fluorescence’ that shows co-localization, but rather overlapping signal of the two fluorescent channels (GFP and Texas-Red) shown in yellow in the merge. Please re-phrase this a bit for scientific accuracy.

Line 39: The authors claim that Aedes aegypti mosquitoes are a vector of SINV without providing a reference. Aedes aegypti are competent vectors for some SINV strains experimentally, but are not generally considered natural vectors. SINV are generally transmitted by Culex and Culiseta mosquitoes. Please re-phrase.

Line 55: contacting? not sure what word the authors meant to use here.

The introduction could include just a little bit more information on the lab’s previous work on foci of infection (reference 18).

Figure 4: The authors present a seemingly random selection of images – Fig 4A has TX-Red FMRFamide stained ECs, and Fig 4D shows a merge of SINV-GFP and TX-Red FMRFamide stained ECs, but not the same image as in ‘A’. Due to the intense signal of GFP, I wanted to see if any co-staining may have been drowned out, so I tried to identify the red stain without merge and noticed that 4A is not the same image/midgut section as 4D. It would be nice to see the GFP, Tx-Red, and merge from the same sample origin next to each other.

In Figure 9, the authors say having foci in all parts of the midgut is an exception, but isn’t that what they also show in Figure 1? In Figure 1 there may be slightly less foci, but one of them also appears larger, so overall it does not look that different from Figure 9... I think to make the claim about this being exceptional, the authors maybe should have provided an average foci count for midguts at 7dpi or similar.

This is minor, but based on ICTV guidelines, a genus, i.e. ‘Alphavirus’, should only be capital and italicized when really used as a genus name. When using the collective name for a group of viruses, i.e. alphavirus/alphaviruses, it should be non-capital and not italicized (e.g. title and line 36).

While this data probably cannot be collected retrospectively for this manuscript, I would urge the authors in the future to include some quantifications/counts that bring the data beyond individual images, similar to Table 1 and what they did in their previous publication (reference 18). For example, in how many mosquitoes was co-localization of SINV and ECs observed? How many of the mosquitoes had a disseminated infection? how many SINV foci were observed on average on day 5 and 7. These sorts of things. As the authors probably know, it is always hard as a reader to evaluate based purely on a few images whether these were cherry-picked or represent the majority of data.

Line 198-200: As mentioned in my major concern, this suggestion is not backed up by the provided data.

Author Response

We thank the Reviewers for your thoughtful comments.  We have thoroughly discussed the comments and believe that we have made changes to the best of our ability and hopefully to your acceptance.  We are sure that this is now a much improved manuscript. 

We have made global changes to Alphavirus to Alphavirus where needed.  We have also discussed Ae. aegypti as a host mosquito not a vector because we have not conducted transmission studies.  Additionally we focused on infection and did not discuss dissemination in terms of our study. Thanks!

Lines 26-31, We have left the title as it was but did change the abstract. 

Lines 133-135 day 5p.i. infection numbers.

Lines 144-146 day 7 p.i. infection numbers.

Line 207 plan to investigate earilier days 2-4 p.i.  Difficult bc bloody until 48 hours post blood meal but we will try our best.   

Lines 229-239 Discussion of low infection numbers and our plans.  

Line 24 reworded in regards to yellow overlap.  Also corrected in figure legends.  

Line 62 changed from contacting to approaching and attaching to SG. (L55)

Lines 88-93 Intro discussion of our previous publication regarding MG foci [20].

Figure 4.

Because this is a whole mount image (somethimes floating), the focal plane can change ever so slightly upon movement of culture dish.  Actually Figures 4A-D are all of the same gut.  If you look very carefully at Fig 4A,B,D there are 2 exoskeleton scales laying left of the foci in a horozontal manner.  These scales are not present in Fig 4C because the gut was rotated to show that the voirus focus extended around the whole mount gut.   So not random at all.  Infact, the graduate student took image 4A then 4B then 4C in that order.  Then took Fig 4C.  

L 90-93 Large number of foci in Figure 9.

L 183-184 High number of foci.

L197-198 Comparison of number of foci in Fig 1 (~6) to Fig 9 (>12).

We changed all the alphavirus/alphaviruses except those listed in literature cited.  Thanks on this one!

L198 is not L207-209  We modified; Ecs are permissive and involved in MG infection.  Also deleated L213-214.  Not overstepping our bounds.  

Round 2

Reviewer 1 Report

.